# Methodology for the Study of Near-Future Changes of Fire Weather Patterns with Emphasis on Archaeological and Protected Touristic Areas in Greece

**Vassiliki Varela \***, **Diamando Vlachogiannis**, **Athanasios Sfetsos**, **Nadia Politi** and **Stelios Karozis**

Environmental Research Laboratory (EREL), INRASTES, NCSR "Demokritos", 15341 Agia-Paraskevi, Greece; mandy@ipta.demokritos.gr (D.V.); ts@ipta.demokritos.gr (A.S.); nadiapol@ipta.demokritos.gr (N.P.); skarozis@ipta.demokritos.gr (S.K.)

\* Correspondence: vvarela@ipta.demokritos.gr; Tel.: +30-210-6503403

**Abstract:** This work introduces a methodology for assessing near-future fire weather pattern changes based on the Canadian Fire Weather Index system components (Fire Weather Index (FWI), Initial Spread Index (ISI), Fire Severity Rating (FSR)), applied in touristic areas in Greece. Four series of daily raster-based datasets for the fire seasons (May–October), concerning a historic (2006 to 2015) and a future climatology period (2036–2045), were created for the areas under consideration, based on high-resolution climate modelling with the Representative Concentration Pathway (RCP), PCR 4.5 and RCP 8.5 scenarios. The climate model data were obtained from the European Coordinated Downscaling Experiment (EURO-CORDEX) climate database and consisted of atmospheric variables as required by the FWI system, at 12.5 km spatial resolution. The final datasets of the abovementioned variables used for the study were processed at 5 km spatial resolution for the domain of interest after applying regridding based on the nearest neighbour interpolating process. Geographic Information Systems (GIS) spatial operations, including spatial statistics and zonal analyses, were applied on the series of the derived daily raster maps in order to provide a number of output thematic layers. Moreover, historic FWI percentile values, which were estimated for Greece in the frame of a past research study of the Environmental Research Laboratory (EREL), were used as reference data for further evaluation of future fire weather changes. The straightforward methodology for the assessment of the evolution of spatial and temporal distribution of Fire weather Danger due to climate change presented herewith is an essential tool for enhancing the knowledge for the decision support process for forest fire prevention, planning and management policies in areas where the fire risk both in terms of fire hazard likelihood and expected impact is quite important due to human presence and cultural prestige, such as archaeological and tourist protected areas.

**Keywords:** fire danger; fire weather patterns; climate change; RCP; FWI system; SSR

## 1. Introduction

Fire plays an important role in ecosystems structure and function in forested and non-forested lands worldwide and in the Mediterranean region as well, where the climate favours noticeable ecological diversity and wildland fire occurrence. All over the Mediterranean areas, fuel moisture level is expected to decrease due to climate change. Thus, the meteorological danger of wildland fires is likely to increase as the region becomes drier [1] with extended low moisture areas northwards [2]. In addition, the presence of human population is intense in many of these areas, making fire hazard and

risk management a major concern and priority. This need emanates from the fact that wildfire activity is projected to increase under future climate conditions and in conjunction with ongoing land use change, forest fires are becoming a reoccurring hazard of forested landscapes globally, posing significant risks to local and regional communities [3,4].

Consequently, new territories start facing increased fire risk but the long term interactions between environmental factors, the social context and the fire regime, as well as the changing fire behaviour spatial patterns, are still largely unknown [5], despite that the relationship of wildfire occurrence to physical and social factors is a widely-researched topic [6,7]. Usually, for the prediction of wildfire occurrence, the factors considered include meteorological data [8,9] and physical indices from fire-danger rating systems [9,10]. Such prediction forms the basis for costly wildfire pre-suppression activities, such as aircraft fire detection flights and pre-fire distribution of firefighting means [11].

The potential for climate change to cause "novel" or "no analogue" environmental conditions in some ecosystems presents new challenges for management, policy and planning [4]. In addition to the day-by-day fire-danger mapping, spatial information related to the fire history and physiognomic characteristics, such as maps of fire frequency, severity, size, and pattern, are useful for planning fire and natural resource management at a strategic level for an area of interest. This type of information can also be used for the assessment of risk and ecological conditions and for the study of fire regimes and their changes as a function of the specific characteristics/features of a territory, such as climate, topography, vegetation and land use [12].

Furthermore, Geographic Information Systems (GIS) can be used for the integration of spatial layers of information for the identification and analysis of spatial patterns of wildfire occurrence and for deriving fire risk at different scales. Different spatial analysis techniques can be applied to answer the questions of "where" and "why" these wildfires are occurring [6,13]. It is also well known that the comparison among regions (e.g., of different geographic orientation and context; northwest, southwest, intermountain region), within regions (across biophysical settings), and across time is a powerful way to understand the factors that determine and constrain the fire patterns [14–16].

It is clearly justified that the main causes of fire have to be minimized, a process which needs to include the investigation of the social and economic factors that lead people to start fires, increasing awareness of the danger, encouraging good behaviour and sanctioning offenders. In particular, the importance of the wildland-urban interface in potentially catalysing fire impacts should be focused on a context where wildfires are genuinely understood as a natural hazard and defensible space is considered even from a social and policy perspective [7].

Montiel et al., 2016 [6], stated that zoning and characterizing a fire-prone territory require special spatial units to analyse the variability of attributes previously defined as relevant to wildfires, both structurally and dynamically. They proposed the use of landscape units obtained from landscape character assessments as the most appropriate ones for a smaller scale approach and drainage basins, which can then be easy to define using GIS, for a larger scale approach.

In addition to the above zoning approach, which is based on physical characteristics, the authors of the current study consider that social and cultural aspects should not be underestimated for the delineation of fire-prone territories and for the definition of respective fire management units. According to Ryan et al., (2012) [17], a landscape approach provides the tools for organizing and understanding intellectual and practical issues engaged by the topic of fire effects on cultural resources.

Any management decisions that affect cultural resources, also affect people and local communities—sometimes in direct and damaging ways. Understanding fuels, fire behaviour, and heat transfer mechanisms is a key to predicting, managing, and monitoring the effects of fire on cultural resources. Cultural resources are important resources that bind those of us living today with our ancestors, traditions, and histories. They are generally viewed as non-renewable resources. They are often fragile tangible objects, susceptible to thermal damage during wildland fires (wildfires and prescribed fires) and physical damage from management-related disturbances [17].

Recognizing these particularities, as well as the necessity of both rational and effective fire management in changing fire regime conditions in Greece due to climate change, the methodology for the study of near-future fire weather changes, proposed in the current paper, focuses on specific areas which cover a number of criteria that combine social and cultural aspects with fire regime and physical characteristics. Greece has suffered from significant forest fires during the last few decades, which, among others, has affected forested archaeological sites [18,19]. As these areas accept thousands of tourists every year, their effective protection and management is essential and imperative.

A number of previous studies elaborated, most of them by using the FWI system indices, on the impact of climate change on the fire regime in the Mediterranean region and particularly in Greece, and indicated that fire danger and risk would increase in the near future in these areas [3,20–22]. Likewise, in a 2011 research work [23], where the FWI system indices and other climatic indices were used with the the Intergovernmental Panel on Climate Change (IPCC) Special Report Emission Scenarios (SRES), A1B, meteorological data and the Regional Atmospheric Climate Model (RACMO2) version 2 of KNMI (Royal Netherlands Meteorological Institute: De Bilt, Netherlands) a substantial increase was found in the number of fire risk days in the near future for a number of areas of agricultural and touristic importance in Greece. Many of the abovementioned studies have used older versions of climatic models, nowadays considered rather less accurate and reliable; however, all the research results have clearly shown the tendency of fire danger and risk to increase in the region, highlighting its importance on economical and societal domains. Moreover, they have put on the table the necessity for studying the fire regime future projection in view of climate change, as an essential prerequisite for fire management, in every area under consideration. Thus, the previous research works mainly focused on the quantitative results obtained for the specific considered geographical region, as well as on the grade of changes in fire danger and risk indices. The current study aims at the provision of indicators and methodological tools for the quantitative assessment of fire weather in "Areas of Interest (AoI)", applicable in any geographical region, that can be considered as distinctive units in terms of all levels of fire management. Accordingly, the AoI selected for Greece, consist of extended areas of the Natura 2000 network, which include archaeological sites or sites of natural beauty with intense touristic load. It is worth mentioning that, according to de Rigo et al., 2017 [5] for specific typologies of forests, increasing the size of protected areas, such as Natura 2000 sites, might even be considered as a potential option for adaptation if other strategies are considered in parallel.

## 2. Materials and Methods

In this section, the developed methodology to obtain the stated objectives of the study are described. The research approach, based on ArcGIS 10.8 (Environmental Systems Research Institute –ESRI: Charlotte, NC, USA), comprised a number of flow processes, which are depicted in the diagram of Figure 1. In a first step, the fire weather calculation for the whole country was carried out, based on the Canadian FWI system components and the input climate datasets of the studied historic and future periods, for the two different emission scenarios. Then, a number of auxiliary thematic layers were selected concerning the country, such as the Natura 2000 map, the bioclimatic map that was part of the ESRI living Atlas and the FWI extreme class thresholds map as derived according to recent past research of the authors [23], Varela et al., 2018. These thematic layers served to select the AoI, taking into consideration a number of criteria, discussed later. In a next step, the selected AoI and the thematic layers were used with GIS functions and tools (e.g., cell by cell analysis, zonal analysis and statistics, spectral profile analysis) to derive the final products which were daily maps and spatial datasets for a number of components of FWI that express the fire spread rate and fire danger. In the sub-sections below, the applied methodology is presented in detail together with the datasets used and the procedure for the analysis of the results.

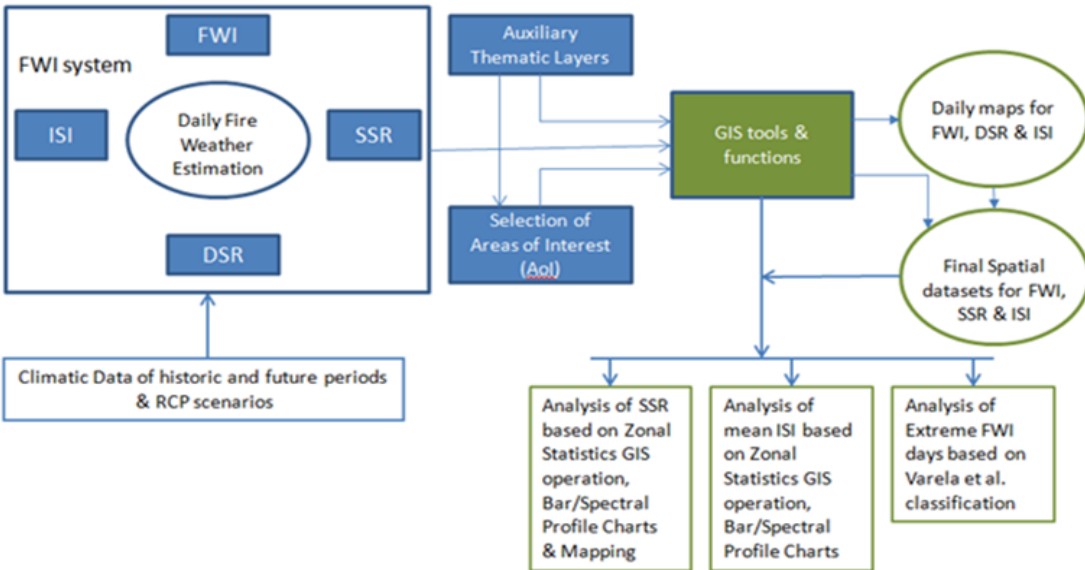

**Figure 1.** Flow process of the developed methodology for the derivation of fire weather patterns for selected Areas of Interest (FWI—Fire Weather Index, ISI—Initial Spread Index, DSR—Daily Severity Rating, SSR—Seasonal Severity Rating, RCP—Representative Concentration Pathway, GIS—Geographic Information System).

### 2.1. Fire Weather Estimation and Climatic Data

The Canadian FWI System, apart from its classical use as a daily fire weather rating system, is widely used for the projection and study of fire weather due to climate change [24,25]. The FWI System is comprised of six components: five of them are intermediate outputs of the system, namely the Fine Fuel Moisture Code, the Duff Moisture Code, the Drought Code, the Initial Spread Index and the BuildUp Index. The final output is the Fire Weather Index (FWI). An extension of the FWI output is the Daily Severity Rating (DSR) [24,25], which is averaged through a fire season for the calculation of the Seasonal Severity Rating (SSR), allowing for the objective comparison of fire danger from year to year and from region to region. It has been used for expressing the projection of Fire danger changes by the European Environment Agency [26].

The parameters of the FWI system that are employed for the purposes of this study are:

- The Initial Spread Index (ISI) which expresses the expected rate of fire spread.
- The Fire Weather Index (FWI) as the main indicator of fire danger representing the potential fire line intensity [14].
- The Daily Severity Rating (DSR) and fire season SSR, which represent the difficulty of controlling fires and reflect the expected fire suppression expected efforts [27].

Daily meteorological values at noon (12:00) of near surface temperature, relative humidity and 10-m wind speed, as well as 24-h cumulative precipitation, are used for the calculation of the components of the system for the whole country. The values of FWI vary from 0 to above 100. The ISI parameter is also used for operational purposes as an indicator of fire spread rate [28]. According to Viegas et al., 1999 [10], ISI was found to be an interesting index for the prediction of extreme fire conditions, as a result of an extended drought and strong wind conditions.

The above parameters of FWI system are classified into 46 classes for operation purposes, using varying classification thresholds, depending on the area of application [23,24,29], and they are an important and effective decision support tool used for forecasting the levels of preparedness needed for an area, the determination of the appropriate mitigation measures, the definition of organizational requirements and the support of fire control measures that are appropriate for the area [24].

In the current study, the climatic data, used as input to the FWI system, were derived from the state-of-the-art Global–Regional Climate modelling system ICHEC-EC-EARTH v2.3 /SMHI-RCA4 v4 (Irish Centre for High-End Computing (ICHEC)-European Consortium-Earth (EC-Earth) version 2.3/Swedish Meteorological and Hydrological Institute (SMHI)-Rossby Centre Atmospheric model version 4) simulations [30] of the openly accessible SMHI archives of the European Coordinated Downscaling Experiment (EURO-CORDEX) at spatial resolution of 12 km. The Regional Climate Model (RCM) simulations were retrieved for the domain of Greece, for the historic (2006 to 2015) and future (2036 to 2045) periods and for the latest IPCC RCP scenarios of climate forcing of 4.5 and 8.5 W/m$^2$ [31,32]. The retrieved variables were values of temperature at 2 m, wind speed at 10 m and relative humidity at 12:00 UTC (Universal Time Coordinated) as well as total precipitation on a 24 h basis.

The RCA4 model performance was evaluated for the recent past climate by Strandberg et al., 2014 [33]. An uncertainty assessment due to systematic bias of the model future climate simulation RCA4 has already been carried out by Sørland et al., 2018 [34]. The excellent performance of the SMHI model has been assessed for Greece by comparing the SMHI-RCA4 EUROCORDEX historic simulations with available meteorological data from the Hellenic National Meteorological Service (HNMS), by Katopodis et al., 2019 [35].

### 2.2. Auxiliary Thematic Layers and Selection of AoI

As previously mentioned, specific thematic layers were used as auxiliary data for the derivation of AoI and the analysis of the results. In particular, the thematic layers employed were the Natura 2000 map for Greece, the Bioclimatic World map and the FWI extreme class thresholds.

The Natura 2000 dataset consists of Special Areas of Conservation (SACs) and Special Protection Areas (SPAs) designated, respectively, under the Habitats Directive and Birds Directive. The Habitats Directive requires Sites of Community Importance (SCIs), which, upon the agreement of the European Commission, become Special Areas of Conservation (SACs) to be designated for species other than birds, and for habitat types (e.g., particular types of forest, grasslands, wetlands, etc.). In Greece, 202 areas have been registered as SPAs and 241 as SCIs. The area covered by the above 443 Greek Natura 2000 areas, cover about 19% of the country [36].

The Bioclimatic World map is part of the ESRI living Atlas [37] and provides access to a 250 m cell-sized raster with a bioclimatic stratification into classes based on factors that influence the distribution of plants and animals [38]. This layer was used to select distribution across the country AoI of various bioclimatic zones reflecting different fire prone areas.

The FWI extreme class thresholds map, according to the methodology of Varela et al., 2018 [23] (Figure 2), was necessary for the analysis and interpretation of the results. According to this methodology, the FWI classification takes into consideration the environmental variety of the country, which highly influences the significance of FWI values and consequently their interpretation as reasonable and functional fire danger classes. The classification approach, applied in the Greek Local Forest Service Office (LFSO) areas, is based on Percentile Indices and provide suitably varying FWI boundaries of classes based on the specific physical characteristics of the study area.

In compliance with the aim of the current study, the AoI were selected in order to satisfy a combination of the following criteria:

- Forest Fire prone areas
- Areas with high cultural and/or touristic interest
- Areas with high ecological/environmental interest
- Distribution in a variety of Bioclimatic zones within Greece

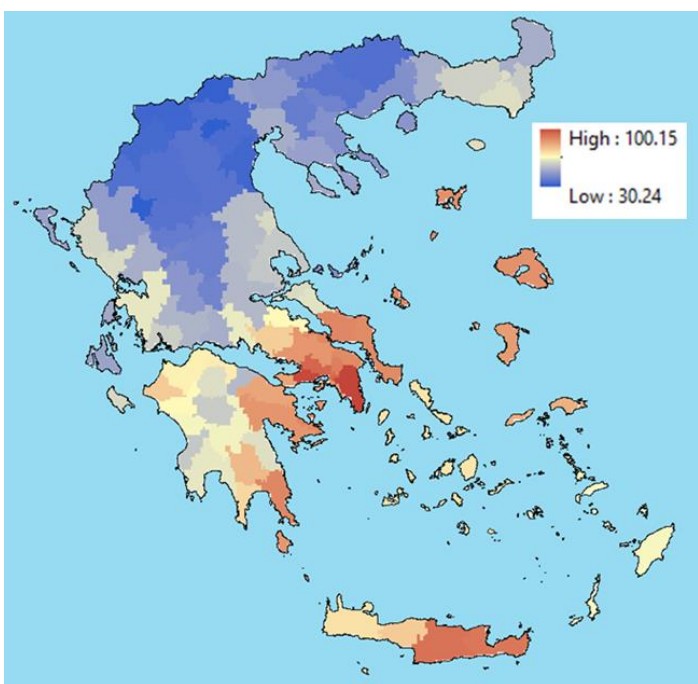

**Figure 2.** Map of Fire Weather Index (FWI) values thresholds for Extreme FWI class for each Forest Office Area.

Thus, the Natura 2000 network of areas was considered as a very important and appropriate basis, since it covers a significant part of the country and includes areas with the above defined criteria. A brief preliminary study of the Natura 2000 areas in conjunction with additional data regarding archaeological sites and touristic load, taking into account the Bioclimatic zones, resulted in the final list of the selected 19 areas (Table 1).

**Table 1.** Catalogue of the selected Natura 2000 areas.

| CODE | HECTARES | NAME |
|---|---|---|
| GR1150012 | 17,592.2 | Thasos (Ypsario Mountain and Seaside Zone) and Koinyra, Xironisi Islands |
| GR2130011 | 53,407.8 | Central Zagori and Eastern Part Of Mitsikeli Mountain |
| GR4210005 | 27,696.2 | Rodos Island: Akramytis, Armenistis, Attavyros, Streams and Seaside Zone (Karavola-Ormos Glyfada) |
| GR1250001 | 19,139.5 | Olympos Mountain |
| GR3000001 | 14,902.4 | Parnitha Mountain |
| GR1270014 | 23,451.1 | Sithonias Peninsula |
| GR4110011 | 14,787.9 | Olympos Lesvou Mountain |
| GR2550009 | 48,785.9 | Taygetos-Lagkada Trypis Mountain |
| GR4210029 | 13,441.9 | Eastern Rodos Island: Profitis Ilias-Epta Piges-Ekvoli Loutani-Katergo, Stream Gadoura-Lindou Stream-Pentanisa and Tetrapolis Islands, Psalidi Hill |
| GR3000013 | 5392.5 | Kythira And Related Islands: Prasonisi, Dragonera, Antidragonera, Avgo, Kapello, Koufo Kai Fidonisi |
| GR4340014 | 13,979.8 | Samaria National Park-Trypitis Canyon-Psilafi-Koustogerako |
| GR2220006 | 20,715.2 | Kefalonia Island: Ainos, Agia Dynati Kai Kalon Oros |
| GR2410002 | 34,384.0 | Parnassos Mountain |
| GR1270003 | 33,567.8 | Athos Peninsula |
| GR1430001 | 31,112.2 | Pilio Mountain and Seaside Zone |
| GR2330004 | 314.8 | Olympia |
| GR2550006 | 53,367.5 | Taygetos Mountain |
| GR3000005 | 5374.3 | Sounio-Patroklou Island and Seaside Zone |
| GR3000006 | 8819.2 | Ymittos-Kaisariani Aesthetical Forest-Vouliagmenis Lake |

For the analytical and methodological purposes of the study, taking into account the extended regime of the Fire Weather influence, as well as the spatial resolution of the climatic data, a buffer of 10 km was applied on the polygons of the selected NATURA areas to create the Areas of Interest (AoI). The buffered areas which comprise the AoI are presented in Figure 3.

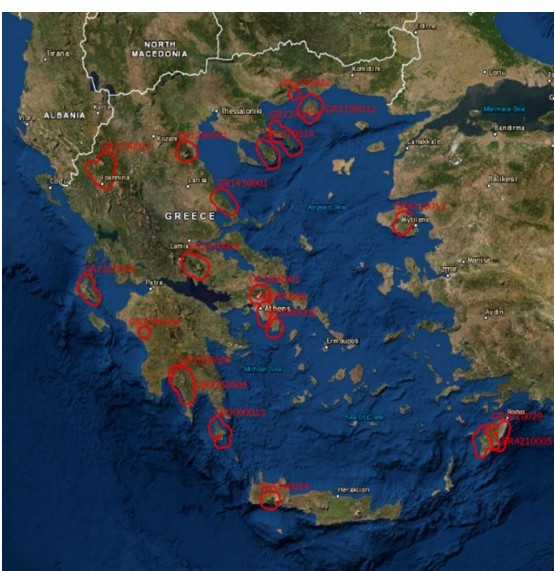

**Figure 3.** Map of the Areas of Interest (selected Natura 2000 areas buffered by 10 km).

## 2.3. Application of GIS Tools and Functions

The study of Fire Weather patterns and changes was elaborated using tools and functions of ARC-GIS v.10.8 software [39].

The software applications, tools and functions adopted for the implementation of the methodology are described below:

i.      FWI System Raster Calculator

For the daily calculation of the FWI system map series, for the historic and future time period the FWI_G.FMIS module of the proprietary software GeographicalFire Management Information System—G.FMIS v.1, (Varela Vassiliki & Eftychidis Georgios, Attika, Greece), has been used [40,41]. The software is developed in C++ programming language, based on the structure and equations of the Canadian FWI system [42] and supports the massive calculation of maps for all the FWI system intermediate and final parameters in ARCGIS Grid ASCII format.

ii.      Buffering

Euclidean buffers measure distance in a two-dimensional Cartesian plane, where straight-line or Euclidean distances are calculated between two points on a flat surface (the Cartesian plane).

iii.      Cell by cell analysis

This function is used for the calculation of per-cell statistics from multiple rasters. The statistics used for the current study are maximum, mean, minimum, range and standard deviation.

iv.      Zonal analyses and statistics

The Zonal tools allow for performing analysis where the output is a result of computations carried out on all cells that belong to each input zone. A zone can be defined as being one single area, but it can also be composed of multiple disconnected elements, or regions.

With the Zonal Statistics tool, a statistic is calculated for each zone defined by a zone dataset, based on values from another dataset (a value raster). A single output value is computed for every zone in the input zone dataset. The Zonal Statistics as a Table tool calculates all—a subset or a single statistic that is valid for the specific input but returns the result as a table instead of an output raster.

v.    Spectral profile analysis

Spectral profile charts allow us to select areas of interest or ground features on the image and review the spectral information of all bands in a chart format [43]. The boxes' chart type of the spectral profile analysis allows us to visualize and compare the distribution and central tendency of the values of the set of pixels for each of the AoI, collected through their quartiles, which are a way of categorizing the examined parameter values into four equal groups based on five key values: minimum, first quartile, median, third quartile, and maximum. A quartile is a type of quantile that divides the number of data points into four more or less equal parts, or quarters. This type of analysis has been used as a supervisory method for the comparison of the distribution and frequency of the values of SSR and ISI parameters within each of the AoI.

In order to study the Fire Weather patterns and their changes in the near future due to climate change, a number of analyses were performed, using the above described GIS functions and operations, for the selected FWI system parameters. The first phase of the work concerned a generic part of application of the FWI system equations for the calculation of daily maps and the creation of the basic spatial datasets. Then, different analyses were considered as appropriate for each parameter, for the provision of easy to interpret and meaningful outputs to be used as quantitative indicators of the fire weather characteristics and their future changes for the areas of interest.

Firstly, the calculation of daily maps for the selected FWI parameters (i.e., FWI, DSR, ISI) was carried out for the historic and future periods for RCP 4.5 and RCP 8.5 scenarios for the AoI, using raster calculation techniques and the algorithm and equations of the FWI system. Then, the FWI system's final spatial datasets were created for the studied periods of time for all the AoI, based on the above daily calculated maps and using the GIS "cell by cell" analyses. More particularly, the groups of spatial datasets that were created concerned:

(a)    The Seasonal Severity Rating (SSR) for the historic and future fire periods for the climatic scenarios RCP 4.5 and RCP 8.5;
(b)    The mean values of Initial Spread Index for the historic and future fire periods for RCP 4.5 and RCP 8.5;
(c)    The number of days with extreme FWI, per fire period for the historic and future fire periods for RCP 4.5 and RCP 8.5. The FWI thresholds for the extreme FWI class for the whole Greece were defined by Varela et.al., 2018 [23].

For each group of the above spatial datasets, i.e., (a–c), further analyses were performed as detailed below.

*2.4. Analyses of SSR Spatial Datasets for the Historic and Future Scenarios*

The Zonal Statistics GIS operation has been used for the calculation of Mean, Maximum, Minimum, STD and Range of values of SSR within each of the AoI for the historic and future period and the respective tables have been calculated. The tables with the calculated values for the historic and the two future period scenarios are presented in Tables S1–S3.

In addition, combined Bar Charts were created for the presentation and comparison of the SSR mean values within each of the AoI for the historic and future period scenarios (Figure 4). Moreover, Spectral Profile Charts of SSR values distribution within each of the AoI, for the historic and future period climatic scenarios were created. Indicative profile charts of two AoI are presented in Figures 5 and 6.

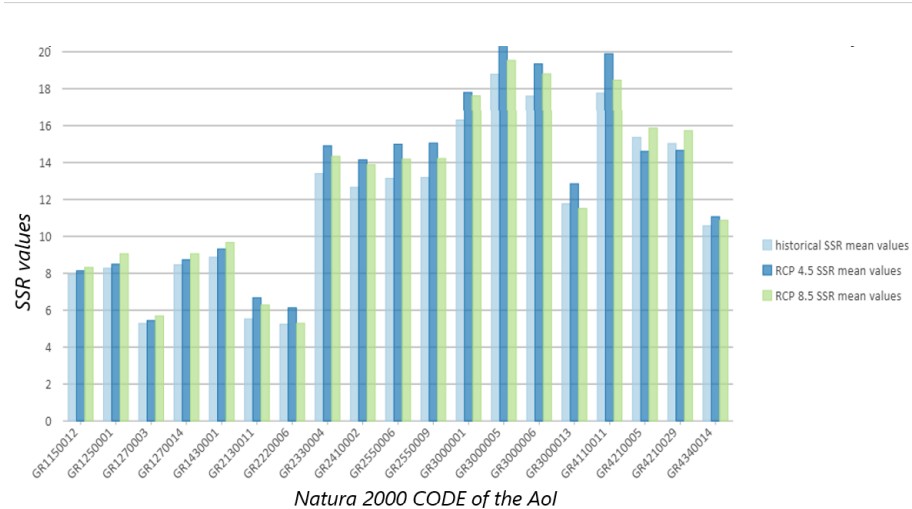

**Figure 4.** Bar chart of historic Seasonal Severity Rating (SSR) value, RCP 4.5 SSR value and RCP 8.5 SSR value by Area of Interest (AoI).

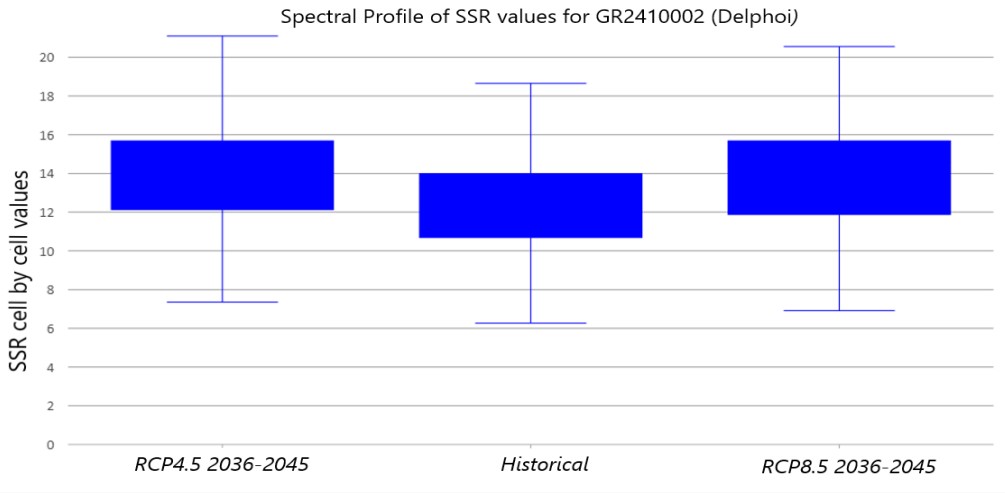

**Figure 5.** Spectral Profile of Seasonal Severity Rating (SSR) values for GR2410002 (Delphoi), for: RCP 4.5 2036–2045 (**left**), historic period (**centre**), and RCP 8.5 2036–2045 (**right**).

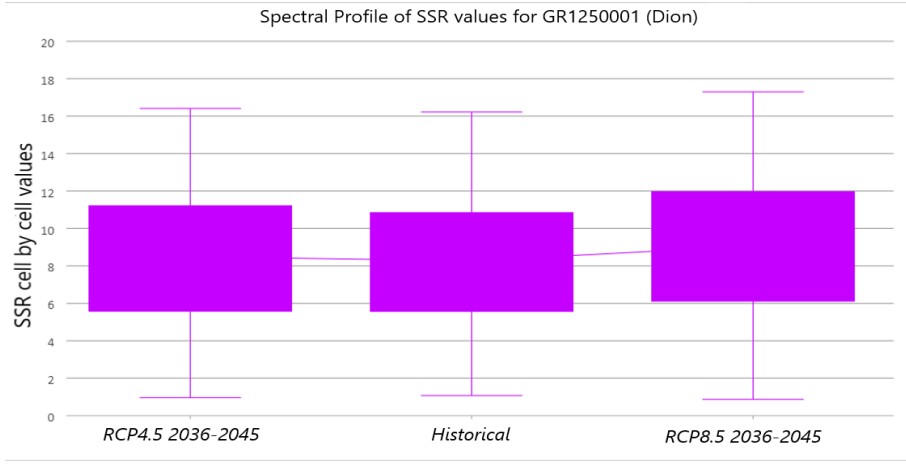

**Figure 6.** Spectral Profile of Seasonal Severity Rating (SSR) values for: GR1250001 (Dion), for RCP 4.5 2036–2045 (**left**), historic period (**centre**), and RCP 8.5 2036–2045 (**right**).

Finally, the mapping of SSR Range and Mean parameters of each of the AoI for the historic and future time-periods, derived from the Zonal Statistics, were classified in categories (Figures 7 and 8). The above selected parameters were considered as essential indicators of Fire weather diversity and intensity for each of the AoI and their mapping provided an informative portrayal and comparison of Fire Weather conditions.

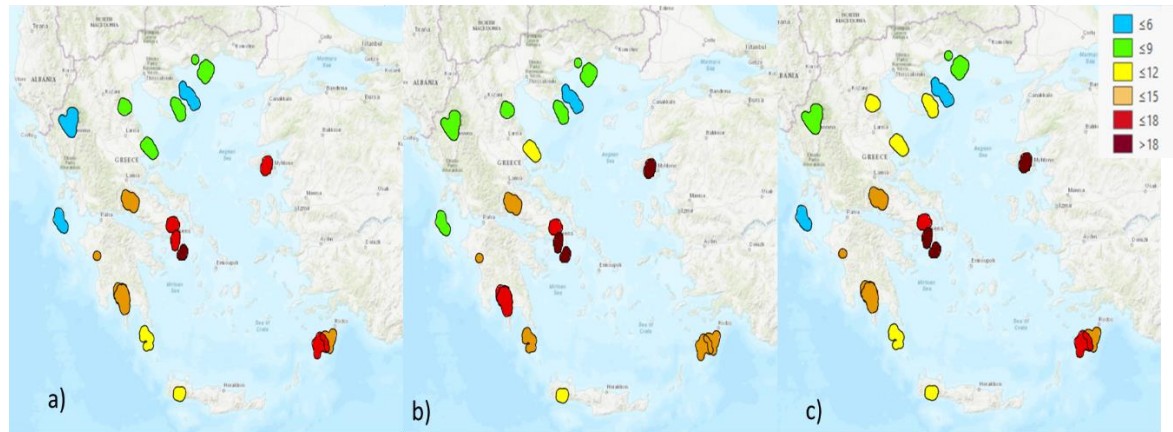

**Figure 7.** Maps of coloured Areas of Interest (AoI) depicting classified Seasonal Severity Rating (SSR) Mean values, for: (**a**) the historic period, (**b**) RCP 4.5 2036–2045, (**c**) RCP 8.5 2036–2045.

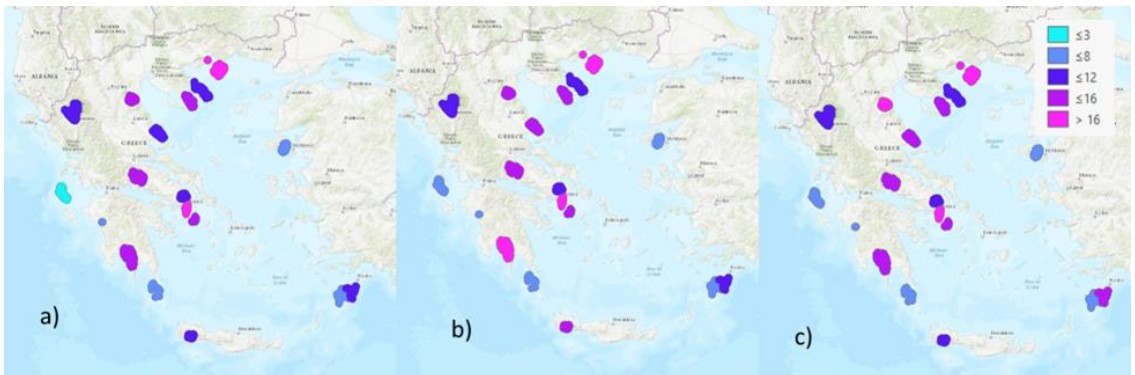

**Figure 8.** Maps of Seasonal Severity Rating (SSR) Range coloured classes within the Areas of Interest (AoI) for: (**a**) the historic period, (**b**) RCP 4.5 2036–2045, (**c**) RCP 8.5 2036–2045.

*2.5. Analyses of Mean ISI Spatial Datasets for the Historic and Future Scenarios*

Similar to the above approach, the Zonal Statistics GIS operation was applied for the calculation of the Mean, Maximum, Minimum STD and Range of values of ISI within each of the AoI for both periods and respective tables have been calculated. The tables with the calculated values for the historic and the two future scenarios are presented in Tables S4–S6.

The combined Bar Charts were then calculated for the presentation and comparison of the mean ISI within each of the AoI, for the historic and future period scenarios (Figure 9). Finally, the Spectral Profile Charts of ISI value distribution within each of the AoI, for the historic and future period scenarios were created. Indicative profile charts of two AoI are shown in Figures 10 and 11.

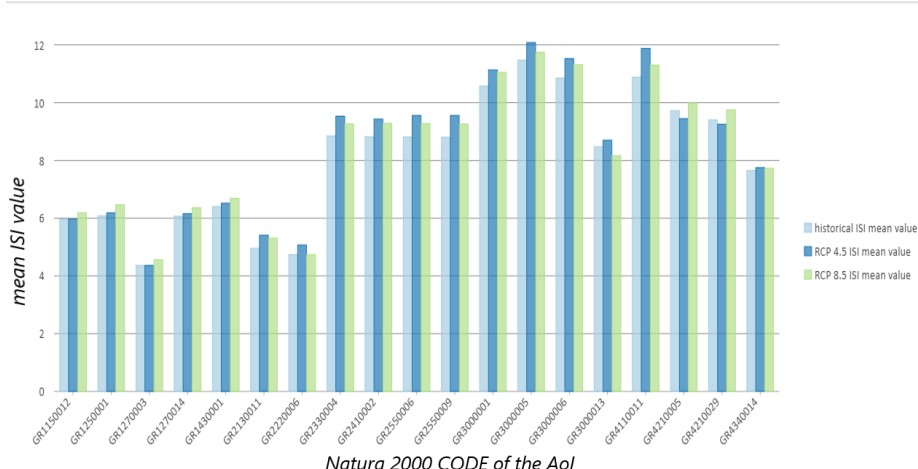

**Figure 9.** Bar chart of historic Initial Spread Index (ISI) Mean value, RCP 4.5 ISI Mean value and RCP 8.5 ISI Mean value by Area of Interest.

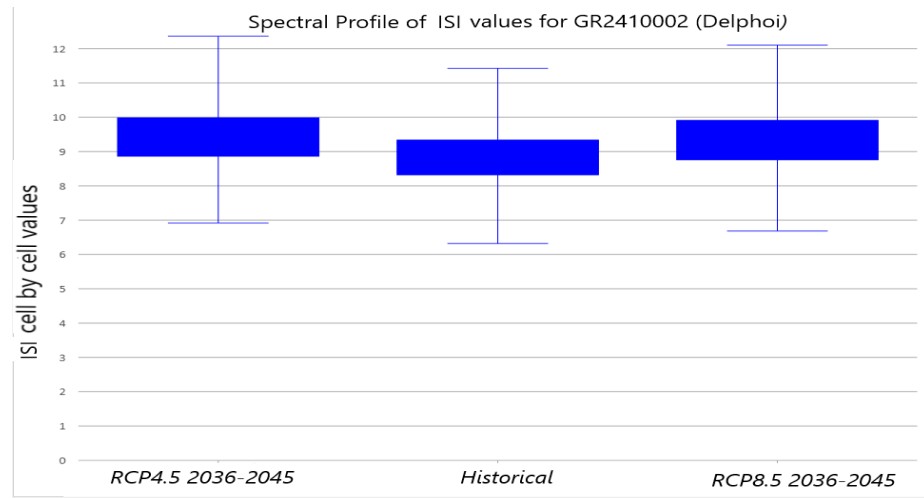

**Figure 10.** Spectral Profile of Initial Spread Index (ISI) values for GR2410002 (Delphoi), for RCP 4.5 2036–2045 (**left**), historic period (**centre**), and RCP 8.5 2036–2045 (**right**).

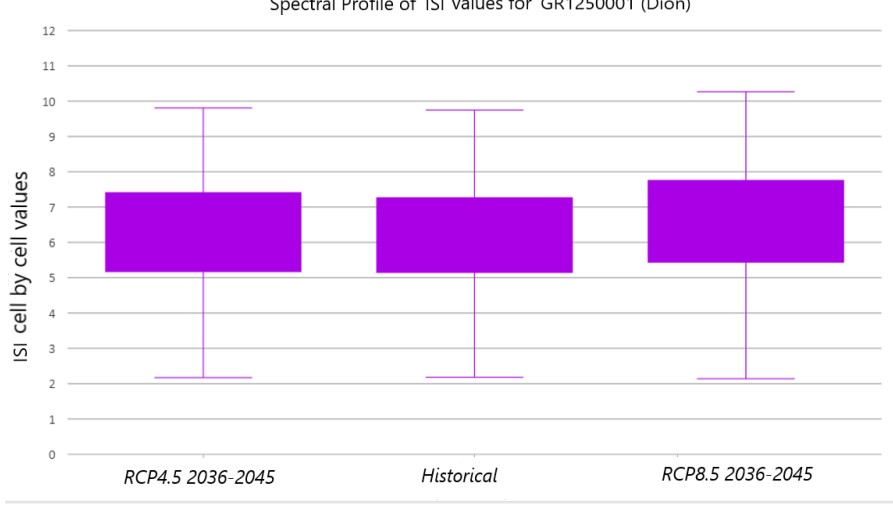

**Figure 11.** Spectral Profile of Initial Spread Index (ISI) values for: GR1250001 (Dion), for RCP 4.5 2036–2045 (**left**), historic period (**centre**), and RCP 8.5 2036–2045 (**right**).

*2.6. Analyses of "Extreme FWI Days" Spatial Datasets for the Historic and Future Scenarios*

The mapping of the difference among the RCP 4.5 and RCP 8.5 future periods and the historic period, of the maximum days with extreme FWI calculated in the cells within each area of interest was performed. Two maps were created as a result of this analysis, each one corresponding to the difference among RCP 4.5 and RCP 8.5 for 2036 to 2045 and the historic period, respectively (Figure 12).

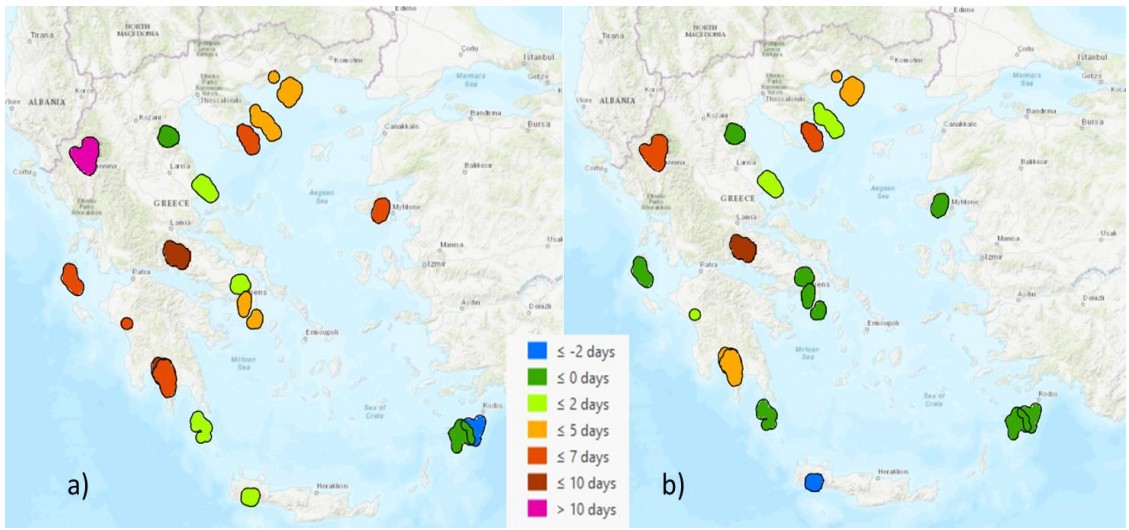

**Figure 12.** Differences in the results of maximum number of days with extreme Fire Weather Index (range depicted with colours) within the areas of interest between: (**a**) RCP 4.5 and historic data sets, (**b**) RCP 8.5 and historic datasets.

## 3. Results

The results of this study, which are based on the methodology described above, are presented in the form of tables, bar charts, box charts and maps, aiming to provide different views of fire weather parameters, in order to allow their study both at a local and national level.

*3.1. Seasonal Severity Rating (SSR) Mapping and Analysis*

The tables with the calculated values derived from Zonal statistics analyses, which are described in Section 2.4 (Tables S1–S3) in Supplementary Material), show that there is a variety of Seasonal Severity Rating (SSR) values for all the statistical parameters, among the AoI for the historic and future periods.

The results indicate that the values of SSR increase everywhere under future scenarios, while this increase is higher for RCP4.5 than for RCP8.5 for a number of areas. However, in three areas, SSR is found to be lower for RCP4.5 than for the historical period. More particularly, the mean values vary from 5.28 for GR1270003 up 18.78 for GR3000005 for the historic period, from 5.44 for GR1270003 up 20.33 for GR3000005 for the future RCP 4.5; and 5.28 for GR2220006 up 20.33 for GR3000005 for RCP 8.5. The range of SSR values within each of the AoI, which shows the diversity of fire weather within the area, is found between 2.79 and 16.39 for the historic period, while for RCP 4.5 and RCP 8.5, SSR values vary from 7.91 to 24.98 and from 6.83 to 24.16, respectively.

Figure 4, which represents the SSR mean values as a bar chart showcases in a more concise manner the variety of the SSR values among the AoI and for the studied periods and scenarios. On the other hand the box plot diagrams in Figures 5 and 6, which are based on Spectral Profile analyses for two (2) selected AoI that were selected as an example for presenting this type of information, provide a more detailed picture of the profile of the SSR values within each of the AoI for both fire periods.

Mapping of the mean and the range of SSR values in Figures 7 and 8, respectively, at the national level, using color-coded classes, allow for visual comparison among the AoI and among the studied time periods. The results show the variety in SSR among the AoI for the historical time, which are

expected due to the diversity of the physical characteristics of the selected AoI, as well as the different levels of changes due to the two future climate scenarios. As an example, in Figure 7, it is obvious that the changes in fire weather due to climate change are more significant for three AoI in northern Greece, which are classified in a higher class and their colour code changes from green to yellow, compared to other AoI, which are classified in the same class as in the historic period map.

### 3.2. Initial Spread Index (ISI) Analysis

ISI parameter is an important intermediate parameter of FWI and similarly to the SSR parameter, Tables S4–S6 (Supplementary Material) and Figures 9–11 provide different analysis views of this parameter for the historical and future periods. The resulting values indicate that the values of ISI also increase in all areas for both future scenarios, while this increase is higher for RCP 4.5 than for RCP 8.5 for most areas.

### 3.3. Difference of the Maximum Number of Days with Extreme Fire Weather Mapping and Analysis

The maps that were derived from the calculation of the difference of the maximum number of days with extreme FWI between the future and historic periods are depicted in Figure 12. The results obtained show quite clearly that, in some areas (i.e., blue, green colour coded AoI), the number of extreme days slightly decreases or does not change significantly, while it is found to increase by more than seven days in other areas (i.e., red, purple colour coded AoI). The greatest difference of more than 10 days is found in the case of RCP 4.5 (Figure 12a).

## 4. Discussion

The analysis described above, is proposed as a methodology for the provision of information for all the areas of interest (AoI), in order to study the various perspectives of fire weather due to climate change.

The resulting values of the fire weather parameters for the selected areas and for the historical and future periods for two climate scenarios of the applied climate model, indicate that significant changes are expected in fire weather due to climate change. Moreover, these changes also have an important spatial variation, which needs to be highlighted, in order to be taken into account for rational fire management adaptation.

The accuracy of the results in terms of the estimated values of fire weather parameters, depends mainly on the inherent uncertainties of the input model climatic data. However, a further analysis on these uncertainties as well on the specific resulting values obtained for the areas under consideration, for the two future scenarios, are considered beyond the scope of the current study, which focuses on the methodology for studying fire weather patterns for distinctive fire management entities. Thus, the results obtained are discussed below mainly from the methodological point of view and the fire management perspective.

Previous research results on climate change impact on forest fires presented the grade of changes by applying a variety of available climate models on different geographical environments. Those studies indicated clearly that climate change should be considered as an important factor affecting future fire regime worldwide and thus as an essential subject for the adaptation of fire management policy and actions [44]. On the other hand, fire management is a complex task, demanding a concise picture of the situation at all operational levels—i.e., at national, sub-national and community (local) levels [45]. The analyses and indicators, which are proposed here, can be applied in local management units and, at the same time, can provide meaningful and easily interpreted results at the sub-national (regional) and national level in a consistent manner. For the presentation and evaluation of the proposed methodology, two climatic models were chosen to be applied in the selected areas in Greece, which were considered as discrete management units. The following discussion of the results aspires to stand out in terms of the operational usefulness of the methodology.

The areas of interest are faced not only as natural protected units but also as landscape entities specifically characterized in terms of Fire Weather, which in turn is the main component of fire danger and risk evaluation for each of the area, at the entity level. Moreover, at a higher level of geographical/administrative analysis (i.e., regional, national), mapping of the classified entities according to specific fire weather related variables, provides a concise portrayal of the spatial distribution and interrelation of fire weather levels at a specific time "instance".

The comparison among the historic and future time "instances", which, in our study, were represented by RCP 4.5 and RCP 8.5 climate change scenarios for the years 2036–2045, permitted the in depth study of the fire weather patterns and their anticipated changes through time. The approach followed included two levels of information, the first covering the needs of an extensive scientific analysis (i.e., tables of variables, graphs) and the other made suitable for direct operational purposes/application (i.e., single and simple Indicators, maps of classified areas of interest).

The basic climatic datasets for the application of the proposed methodology were based on two IPCC RCP scenarios of 4.5 and 8.5 W/m$^2$, from the openly accessible SMHI-RCA4 archives. The data were spatially downscaled from 12 to 5 km resolution by applying a regridding method based on the nearest neighbour interpolating process.

The zonal statistics for Seasonal Severity Rating (SSR) and Initial Spread Index (ISI) parameters are presented in Tables S1–S6 (Supplementary material). The Range of Seasonal Severity Range (SSR) and Initial Spread Index (ISI) values within each area of interest, as well as their Mean value and Standard Deviation (STD) for the historic and the future scenarios were the descriptive indicators of these two important parameters of fire weather. The calculated indicators allowed for both the comparison among the areas for a specific time lapse and the study of fire weather evolution in time. From the Range and STD values in these Tables S1–S6, it was deduced that some areas were characterized by intrinsic homogeneity in both parameters (e.g., GR2220006, GR23300004) while others showed high diversity in fire weather levels (e.g., GR1150012, GR 1250011).

The bar charts analyses of the Mean values of both parameters (SSR and ISI) for the areas of interest, which were shown in Figures 4 and 9, respectively, constituted a comprehensive means of presentation and comparison of the level of historic and future fire weather for the areas. According to this analysis, both future scenarios lead to an increase in the two parameters in all the areas under examination.

Furthermore, spectral profiles of each of the area of interest (Figures 5, 6, 10 and 11) could provide additional information about the distribution of mean and extreme values of parameters within each of the area for all the time periods, and could be used in conjunction with the bar charts.

This type of information, which is available at the "area of interest" level of analysis, can be useful as a decision support background for the fire management actions of the agency that is responsible for the area (i.e., local forest office, management body), as it provides a brief overview of the current and future fire weather conditions within the area. In addition to the above outputs, classification and mapping of the areas of interest according to historic and future SSR Mean and SSR Range (Figures 7 and 8), can provide a comprehensive and informative view for operational purposes at a regional or national level. It is worth mentioning at this point that, for the classification thresholds and the number of classes for SSR and ISI parameters, further research is essential for the definition of appropriate values, customized at a regional or national level.

The classification thresholds of these two indicators, for the purposes of the current paper, were defined based on the range and distribution of values that were obtained for the examined areas in the historic period in a way to accommodate a satisfactory for the analysis number of classes. The classification of the mean SSR values lead to a map of six classes of areas of interest. For the future scenarios, the classes were found to change for some of the areas of interest. More particularly, for RCP4.5 in total six areas of interest were found at a higher class and two areas of interest at a lower class, while for RCP 8.5, seven areas of interest were shown at a higher class.

Similarly, the classification for the SSR range of the historic period lead to five classes of areas of interest. For the future RCP 4.5 scenario, three areas changed to a higher class and one areas to a lower class, while for RCP 8.8, four areas were found at a higher class.

The estimation and mapping of the difference of the maximum number of days with extreme fire weather between historic and future periods (Figure 12) is another informative output of the proposed methodology, to be used for operational purposes, as it is a valid indicator of the potential changes of the preparedness planning actions.

Further information can be figured out about the patterns of changes. For example, some areas that are characterized by humid and cold climate (e.g., GR21300011, GR2410002) are expected to suffer significant changes in the number of extreme fire weather days, while others which belong to the classical Mediterranean zone show very slight changes (e.g., GR3000013, GR43400014). This is considered an interesting finding, since it indicates that climate change tends to affect, more severely, areas less adapted to forest fires. In those areas, ecosystems are less resilient on wildland fires and the recovery after a fire occurrence will take much longer or even may not occur at all. Future research on specific characteristics of the areas in terms of the climatic zone and other physical descriptors may provide interesting conclusions about the fire regime due to climate change aspects.

The above output indicators and physical descriptors of the methodology presented as maps and tables for the selected areas in Greece, are easily applicable to other geographical areas, either at a local or at a national level, since they are based on classical GIS functions. Besides, the mapping of FWI system parameters for the study of current and future fire weather, which constitutes the underlying dataset for the development and application of the methodology in any geographical area, is a common practice for all the categories of relevance to forest fire stakeholders. Moreover, as implied above, the introduced methodology can be the basis for further enhancement of indicators and descriptors related to forest fire regimes, aiming to facilitate fire management to a greater extent.

## 5. Conclusions

A straight forward methodology was presented for the estimation of fire weather indicators of the current fire weather and for the near future for areas which are considered as management units in terms of fire management. The proposed methodology, easy to apply using simple GIS functionality, can be used as an informative decision support tool for operational purposes in any geographical area, at a local and national level. The application of the methodology provided interesting, easy to interpret results for the area of Greece for the anticipated changes in fire weather danger, due to near future climate changes. The outcomes of the methodology, which are provided both as indicators for individual areas and as maps at a regional or national level, could be used in conjunction with other thematic layers and information for the areas of interest. The combination and mapping of the various indicators provide a concise view of the fire weather characterizing each area and also allow for comparison among the management units and the extraction and study of spatiotemporal patterns. This in turn constitutes a valid approach for providing in depth knowledge of the current and future fire weather regime which would not be obvious otherwise.

**Supplementary Materials:** The following are available online at http://www.mdpi.com/1999-4907/11/11/1168/s1, Table S1: Statistics for the areas of interest for SSR calculated for the historic dataset, Table S2: Statistics for the Areas of interest for SSR calculated for the scenario RCP 4.5 for the period 2036–2045, Table S3: Statistics for the Areas of interest for SSR calculated for the scenario RCP 8.5 for the period 2036–2045, Table S4: Statistics for the AoI for mean ISI cell values calculated for the historic dataset, Table S5: Statistics for the AoI for mean ISI cell values calculated for the scenario RCP 4.5 for the period 2036–2045, Table S6: Statistics for the AoI for mean ISI cell values calculated for the scenario RCP 8.5 for the period 2036–2045.

**Author Contributions:** Author Contributions: Individual contributions for this paper were as follows: conceptualization, V.V.; methodology, V.V. and D.V.; software, V.V. and S.K.; validation, V.V., A.S. and D.V.; formal analysis, V.V., D.V. and A.S.; investigation, V.V. and N.P.; resources, V.V., A.S., N.P.; data curation, V.V. and S.K.; writing—original draft preparation, V.V.; writing—review and editing, D.V.; visualization, V.V.; supervision, V.V. and D.V.; project administration, A.S.; funding acquisition, A.S. All authors have read and agreed to the published version of the manuscript.

**Funding:** This research was funded by project XENIOS T1EΔK-02219 of the Greek Operational Programme Competitiveness, Entrepreneurship and Innovation (EPAnEK) of NSRF 2014-2020

**Conflicts of Interest:** The authors declare no conflict of interest.

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
