# Peer review of "Methodology for the Study of Near-Future Changes of Fire Weather Patterns with Emphasis on Archaeological and Protected Touristic Areas in Greece"

_forests, doi:10.3390/f11111168_

Round 1

Reviewer 1 Report

The manuscript is interesting, well written and scientifically robust. 

It introduces an effective and practical methodology, detailed enough so that  that others can replicate it. In some points where they do not describe steps in detail, they provide the relevant citations.

There are no methodology problems I can identify but there are minor problems in English. I have introduced proposals for corrections with sticky notes in the attached file.

The one weakness that could be improved is that the authors have not made a strong effort to refer to other research publications that have addressed this topic. This probably makes the introduction too succinct. Most important, it would be desirable to introduce comparisons of the findings to those of other studies.

Author Response

Dear  Reviewer,

Thank you very much for your comments and suggestions for the improvement of our manuscript. Please see the attachment for our detailed response in every point of your suggestions.

Yours faithfully,

Vassiliki Varela

On behalf of all the authors

Reviewer 2 Report

The manuscript entitled ”Methodology for the study of near-future changes of Fire Weather Patterns with emphasis on archaeological and protected touristic areas in Greece” is clear, well structured and well-written. The objective is properly introduced and justifies the importance of this study. This paper is a new and original contribution on methodology for assessing near-future fire weather patterns changes based on the Canadian FWI system which improves current methods.

The manuscript fits to the scope of the journal, title reflects the content of manuscript well and abstract is informative enough. The introduction is clear and introduces well the specific objective.

Taking into account that the aim of the study is to propose a new methodology, this chapter is detailed.

Results from this study will be of interest to fire and land management agencies, especially considering that more wildfires are projected under the current climate change condition. Nevertheless, in my opinion there are too many Tables, and Figures giving the same information. I would recommend to move some of the tables to “Suplemmentary Material”.

The discussion section is quite vague and basically it is a repetition of the Introduction. I believe that the author should discuss advantages and disadvantages of the proposed methodology, limitations and the feasibility to be replicated in other parts of the world, especially in countries of the Mediterranean Basin. Also, they should discuss the obtained results in the different selected areas.

In conclusion, it is a good study that will provide important contribution to fire and land managers, but it needs some corrections.

Specifically, the changes needed are:

Line 223-229 Delete. It is a well known concept.

Author Response

(The authors gave the same response as above.)

Reviewer 3 Report

The title of the paper corresponds to the content analyzed in the manuscript. Selected methods scientifically present the set issues. Graphic representations clearly show the settings and research results.

Author Response

Dear Reviewer,

Thank you very much for your review and comments about our manuscript. 

Yours faithfully,

Vassiliki Varela

On behalf of all the authors

Round 2

Reviewer 2 Report

The manuscript has been improved according the recommendations

Author Response

Dear Reviewer, 

We would like to thank you for the review of our manuscript and for helping us to improve it for publication.

Yours faithfully

Vassiliki Varela

on behalf of all the authors